**Data Availability Statement:** All relevant data are within the manuscript and its Supporting Information files.

# Relationship of sociodemographic and lifestyle factors and diet habits with metabolic syndrome (MetS) among three ethnic groups of the Malaysian population

**Saleem Perwaiz Iqbal** *, **Amutha Ramadas, Quek Kia Fatt, Ho Loon Shin, Wong Yin Onn, Khalid Abdul Kadir**

Jeffrey Cheah School of Medicine and Health Sciences, Monash University, Bandar Sunway, Malaysia

* saleem.iqbal@monash.edu

## Abstract

### Objectives

Literature shows a high prevalence of MetS among Malaysians, varying across the major ethnicities. Since sociodemographic characteristics, lifestyle factors and diet habits of such communities have been reported to be diverse, the objective of this study was to investigate the association of various sociodemographic characteristics, lifestyle factors and diet habits with MetS overall, as well as with the three major ethnic communities in Malaysia, specifically.

### Materials and methods

We conducted a cross-sectional study among 481 Malaysians of ages 18 years and above living in the state of Johor, Malaysia. Information on demographics, lifestyle and diet habits were collected using a structured questionnaire. Harmonized criteria were used to assess the status of MetS. Multiple logistic regression was employed to determine any associations between sociodemographic and lifestyle factors and dietary behaviours with MetS.

### Results

MetS was found among 32.2% of the respondents and was more prevalent among the Indians (51.9%), followed by the Malays (36.7%) and the Chinese (20.2%). Overall, increasing age (AOR = 2.44[95%CI = 1.27–4.70] at 40–49 years vs. AOR = 4.14[95%CI = 1.97–8.69] at 60 years and above) and Indian ethnicity (AOR = 1.95[95%CI = 1.12–3.38)] increased the odds of MetS, while higher education (AOR = 0.44[95%CI = 0.20–0.94] decreased the odds of MetS in this population. Quick finishing of meals (AOR = 2.17[95%CI = 1.02–4.60]) and low physical activity (AOR = 4.76[95%CI = 1.49–15.26]) were associated with increased odds of MetS among the Malays and the Chinese, respectively.

### Conclusion

The population of Johor depicts a diverse lifestyle and diet behaviour, and some of these factors are associated with MetS in certain ethnic groups. In the light of such differences,

**Funding:** Financial assistance was provided by the Clinical Research Center, Monash University, Malaysia, and by the grant awarded to AR by the Malaysian Ministry of Higher Education's Fundamental Research Grant Scheme (FRGS/2/2013/SKK07/MUSM/03/1). The granting agency had no role in study design, data collection and analysis, decision to publish, or preparation of the manuscript.

**Competing interests:** The authors have declared that no competing interests exist.

ethnic specific measures would be needed to reduce the prevalence of MetS among those in this population.

## Introduction

Metabolic syndrome (MetS) is a combination of interrelated risk factors that predispose individuals to the development of cardiovascular disease (CVD) and diabetes. This includes hyperglycemia, raised blood pressure, hypertriglyceridemia, low high-density lipoprotein (HDL) cholesterol levels, and abdominal obesity, and is now recognized as a disease by the World Health Organization (WHO) and other international entities [1, 2].

According to a study done across seven European countries, the overall prevalence of MetS was estimated to be 23% using the WHO criteria [3]. In Canada, nearly 25% of the adult population was found to be afflicted with MetS using the National Cholesterol Education Program (NCEP)–Adult Treatment Panel III (ATP III) [4]. In Australia, the prevalence values of MetS using the WHO, NCEP-ATP III and International Diabetes Federation (IDF) criteria were 21.7%, 21.1% and 30.7%, respectively [5, 6]. This points to the fact that prevalence of MetS within the same region may vary according to the definitions employed. Moreover, this variation could be due to the differences in the defined cut offs for its associated metabolic components.

As the proportion and distribution of body fat in Asians, in general, was found to be different from the populations in North America and Europe, it became apparent that the definition of obesity applied to Western populations was not be applicable to Asian populations [7, 8]. Therefore, the estimated prevalence values of MetS among Asians were found to be increased when Asian-adapted definitions of obesity were employed in the NCEP-ATP III. For example, in the Southeast Asian region, it increased from 13.1% to 20.9% for Singaporean males, and for the Chinese adults, it increased from 10.1% to 26.3% [9, 10]. A similar trend was observed among the Malaysians where during the 2008 nationwide survey an overall prevalence of 42.5% from 4341 subjects was reported using the Joint Interim Statement (JIS) "Harmonized" criteria, compared to 34.3% via the NCEP-ATP III criteria [11]. Like in NCEP-ATP III, MetS according to the Harmonized definition includes any three of the five metabolic abnormalities–central obesity, hypertriglyceridemia, low HDL-cholesterol, high blood pressure and hyperglycemia [2]. However, the Harmonized criteria have defined Asian cut-offs for central obesity (waist circumference: $\geq$ 90 cm for males; $\geq$ 80 cm for females) and reduced cut-off for hyperglycemia ($\geq$ 5.6 mmol/L, instead of 6.1 mmol/L in the NCEP-ATP III). Ramli et al., using the Harmonized definition reported the prevalence of MetS to be 43.4% in 2013 among 8,836 subjects across East and West Malaysia [12]. This percentage was very close to the 42.5% prevalence reported in the 2008 nationwide survey [11].

The prevalence of MetS is dependent on a variety of non-modifiable (gender, age, ethnicity) and modifiable (lifestyle, diet) risk factors. These factors are known to, directly or indirectly, influence MetS among the populations. For instance, Wen and colleagues reported the prevalence of MetS in rural China as 44.3% (by modified NCEP-ATP III criteria), 40.7% (by IDF criteria) and 47.7% (by Harmonized criteria), amongst a large cohort of 4748 subjects, primarily females aged 50 years and above [13]. From a study in Canada, Liu and colleagues reported MetS prevalence to be higher among the Cree Indians compared to other aboriginal and non-aboriginal Canadians [14]. Similarly, the prevalence of MetS in Malaysia is not different, as according to the nationwide survey in 2008, the prevalence was found to be higher among older age groups, more among females, and most common among the Indians compared to other races in Malaysia [11].

Studies have shown that various lifestyle factors influence MetS. A sedentary lifestyle and physical inactivity are factors that have been shown to contribute to the development of MetS and its components [15–19]. Smoking and alcohol consumption have also shown to have variable influences on MetS and its components [20–24]. Furthermore, diet habits such as speed of eating, dining out, skipping breakfast and late dinners have been found to be associated with increased incidence of MetS [25, 26]. These factors are present in most ethnic communities and might provide some insight into how their influence on MetS could be regulated among populations to contain its life-threatening complications.

Reports mentioned above indicate the significant influence of lifestyle habits on the prevalence of MetS in a particular population. Malaysia is a unique country in Southeast Asia because of its ethnic diversity, culture, lifestyle choices and dietary intake habits. The influence of differing lifestyle choices and diet habits across the three major races of the country may provide a better understanding of the high prevalence of MetS in the country, along with measures for its containment. There have been very few studies carried out in Malaysia on investigating the influence of lifestyle factors with the risk of MetS among the Malaysian population [27–29]. While the two studies by Chu and Moy described the influence of physical activity on MetS among the Malays, the only study which dealt with ethnic differences with respect to physical activity and prevalence of MetS among the Malaysian population was based on the data that were collected more than 13 years ago [27–29]. Moreover, in that study the relationship of lifestyle behaviors with MetS among major ethnicities was not reported [29]. Therefore, the objective of the present study was to determine the association of sociodemographic characteristics, lifestyle factors and diet habits with the risk of MetS, overall and among the three major ethnic groups residing in Johor, Malaysia.

## Materials and methods

### Study design and location

This was a cross-sectional study, employing a nonprobability sampling strategy, conducted in Kulai (in May 2016) and Felda Taib Andak (in August 2016) of the Kulai district and Johor Bahru (in December 2018), Ulu Tiram (in February 2017) and Kota Masai (in July 2017) of the Johor Bahru district of Malaysia. Based on the Department of Statistics Malaysia, Johor represents the other states of Malaysia in terms of ethnic distribution. Furthermore, the selection of the above-mentioned study locations was based on the available percentages of MetS across each major ethnicity of Malaysia, reported in the nationwide survey 2008 [11]. This was to have enough subjects to represent each ethnicity so that data could be available for in-depth analysis for the stated objectives. As a result, the selected sample size from these locations was considered to be generalizable to some extent to the population of Johor, and to population of some of the other states in the country, with similar distribution of the three major ethnicities.

### Recruitment and eligibility criteria

Research camps were set up in central locations of Kulai and Johor Bahru districts, which were easily accessible to the target community. Assistance was sought from community elders for making the locals aware of the research camps and to convey our requests for their participation.

The inclusion criteria for the study participants were that the subjects should be of ages 18 years or above, of either gender, and had been residing in Johor for at least one year. The subjects were requested to observe a 10 to 12-hour fast before arriving at the medical camp to donate blood samples for accurate assessment of fasting serum levels of glucose, triglycerides and HDL-cholesterol. Exclusion criteria included pregnancy or having any illness which could

preclude their participation in the study such as cancer, liver disease, etc. Consented participants were invited to visit these camps for a physical examination and collection of fasting blood samples. After sample collection, the subjects were asked about their lifestyle and dietary habits. Participants who did not observe the 10–12 hour fast were excluded from the analysis.

### Data collection and measurement

Data from the participants were collected using a structured questionnaire and a proforma which contained information on anthropometric measurements, measurement of blood pressure, blood sample analysis results and questions on sociodemographic, lifestyle factors and diet behaviors. The questionnaire was designed in English and back translated into the Malay language. In the study, the Malay language version was used. The questionnaire was pretested on a sample of 29 subjects in Kulai before its employment on the main target population. A copy of the questionnaire is attached as supporting information (S1 File).

Body height was measured using Seca stadiometers (Seca, USA), while the weight was measured using the InBody 120 body fat analyzer (Biospace, Korea). Steps were taken to ensure the subjects wore light clothing and had no shoes on. The measurement was recorded to the nearest 0.1 cm and 0.1 kg, respectively.

Waist circumference was measured using a measuring tape. The measurements were taken at the mid-point, between the lower rib margin (12th rib) and the iliac crest. Caution was taken during measurements that the subject was standing straight with feet together and arms relaxed on either side. Furthermore, it was ensured that the tape was held in a horizontal position, wrapped around the waist, loose enough for the assessor to insert his/her finger between the tape and the subject's body. The subject was instructed to breathe normally during the assessment, with the measurement recorded at the end of a normal exhalation and rounded to the nearest 0.1 cm.

Blood pressure was recorded using the Omron digital sphygmomanometers (HEM-7121, Omron Healthcare, Japan). The subject was provided a 4–5 minute rest, in a seated position, with the arm supported at heart level. At least two readings were taken from each subject, recording the concurrent or highest measurement obtained from the two readings. A third reading was taken in case, the difference between the two readings for the systolic blood pressure was more than 10 mmHg, and for the diastolic blood pressure more than 5 mmHg.

Fasting blood samples were collected from the study participants for determining the levels of fasting serum glucose (in mmol/L), fasting serum triglycerides (in mmol/L) and fasting serum HDL-cholesterol (in mmol/L). Standard guidelines for phlebotomy were followed throughout the venepuncture procedure [30].

Collected samples were transported in cold chain to the laboratory where these were centrifuged, and the sera samples were separated and placed in identity marked cryotubes or Eppendorf tubes. These were then placed in a -60 degree Celsius freezer till laboratory analysis.

The blood analysis for determination of serum levels of fasting glucose (mmol/L), triglycerides (mmol/L) and HDL-cholesterol (mmol/L) was carried out using clinical chemistry analyser (Cobas C III). Its reagents were purchased from Randox Laboratories, United Kingdom.

The participants' physical activity status was determined using the validated International Physical Activity Questionnaire (IPAQ) [31]. The questionnaire comprised seven questions; the first two pertaining to the time spent on vigorous activities performed, the next two for moderate activities, the next two for mild activities and the last question was on the time spent. Responses were converted to Metabolic Equivalent Task minutes per week (MET-min/week) according to the IPAQ scoring protocol. The protocol also provides details for data processing, cleaning and truncation. The total minutes spent on vigorous, moderate, and mild activities

over the last seven days were multiplied by 8.0, 4.0, and 3.3, respectively, to create MET scores for each activity level. MET scores across the three sub-components were then summed up to indicate the overall physical activity score. These overall scores were then categorized into high (total activity of at least 3000 MET-min/week), moderate (total activity of at least 600 MET-min/week) and low (total activity < 600 MET-min/week) level activities.

Diet habits included quick finishing of meals, frequency of late dining, frequency of skipping breakfast and frequency of dining out. For quick finishing of meals, the question was asked on the subject's perception on finishing their meals either fast (less than 10 minutes) or not fast [26, 32, 33]. The assessment of the other three diet habit questions (frequency of late dining, frequency of skipping breakfast and frequency of dining out) were based on the participant's frequency per week; three times or less were considered favorable [26]. "Late dining" was defined as a meal eaten within two hours before bed-time. "Dining out" was defined as a meal consumed by the participant that is not prepared at his/her home [34–36].

## Definition of MetS

MetS was defined using the Harmonized criteria as having at least three of the following five risk factors: 1) Abdominal obesity, defined as having a waist circumference ≥ 90 cm for males and ≥ 80 cm for females; 2) Raised serum triglycerides (hypertriglyceridemia), defined as ≥ 1.7 mmol/L (150 mg/dL); 3) Low high density lipoprotein cholesterol (HDL-C), defined as < 1.0 mmol/L (40 mg/dL) for males and < 1.3 mmol/L (50 mg/dL) for females; 4) Raised blood pressure, defined as a systolic blood pressure ≥ 130 or a diastolic blood pressure ≥ 85 mmHg, or current use of anti-hypertensive medications; and 5) Raised fasting blood sugar (hyperglycemia), defined as ≥ 5.6 mmol/L (100 mg/dL) or current use of anti-diabetic medications.

## Statistical analysis

Data entry was performed using EpiData version 3.1. During the process of data entry, 5% of the forms were re-checked for accounting of any errors during entry of data. All data were analyzed using Statistical Package for Social Sciences (SPSS) version 23 (IBM SPSS Statistics for Windows, Version 23.0. Armonk, NY: IBM Corp.).

The sample size estimate was calculated using estimates of various components of MetS reported in the 2008 nationwide survey [11]. According to the calculation, increased blood pressure (≥ 130/85 mmHg) yielded the sample size estimate of 386 at 5% level of significance and precision of 0.05.

Frequencies and percentages were obtained for categorical variables. Chi square tests for Independence were used to determine the univariate association between categorical variables. Multiple logistic regression analyses were used to determine the associations of sociodemographic and lifestyle factors with MetS, calculating odds ratios with 95% confidence intervals, while adjusting for confounding factors. Variables, with $p < 0.25$ on univariate analysis were selected for adjustment in the final logistic regression model. A $p < 0.05$ was considered statistically significant.

## Ethics

Ethical approval was sought from the Monash University Human Research Ethics Committee (Project # CF15/56-2016000022), which was granted before the start of sample collection.

## Results

The prevalence of MetS was found to be 32.2% in the study subjects, according to the Harmonized criteria; highest among the Indians (51.9%), followed by the Malays (36.7%), and lowest among the Chinese (20.2%) (Fig 1). Abdominal obesity (62.0%) and high blood pressure (56.8%) were more common compared to other metabolic abnormalities. Three most prominent MetS risk factors among the Malays and the Indians were abdominal obesity, high blood pressure and low HDL-cholesterol. Among the Indians however, the percentages of abdominal obesity and HDL-cholesterol were higher than that among the Malays. Prevalence of high blood pressure was more prominent among the Malays compared to the other ethnic groups. Among the Chinese, the third most prevalent risk factor was hypertriglyceridemia. Prevalence of low HDL-cholesterol was lowest among the Chinese.

Table 1 shows the summary statistics of sociodemographic, lifestyle and diet characteristics overall, and across the 3 major ethnicities in Johor. The target population consisted of 64.9% females, while the remaining were males. About 49.9% of the population were of ages 50 years and above; from which 25.6% of them were aged 60 years and above. A majority of the participants were married (78.6%), attaining at least secondary education (60.5%) and were unemployed (57.2%). Thirty-one percent of the target population were Malays; Chinese were 47.4% while the Indians constituted 22% of the target population.

Table 1 also shows the comparative association of sociodemographic, lifestyle and diet characteristics with MetS overall, and across the three major ethnic groups in Johor. Overall, significant differences were observed with age, ethnicity, marital status, education and physical activity (p < 0.05). Marital status and education were found to be related significantly with MetS among the Malays, while age and physical activity among the Chinese and age among the Indians showed a significant association with MetS. Among the Malays, 59.4% of the people with primary education or lower were having MetS, suggesting that the Malays having higher education appear to be protected against the risk of MetS (p < 0.05). A majority of the Indians appear to be afflicted with MetS at a younger age (41.8% at the age group of 40–49 years; p < 0.001). Conversely, only 15.2% of the Chinese were suffering from this syndrome in this age group (p = 0.016). This shows that the Chinese in Johor are getting this disease at a relatively older age.

Table 2 shows the adjusted multiple logistic regression model, the results of which indicate that overall in this population, higher age groups and the Malaysian Indians had increased odds of MetS, while the Chinese ethnic group and those with tertiary education were protected against the risk of MetS. Lifestyle factors and diet habits did not appear to have any association with MetS, overall, in the adjusted model (p > 0.05).

In view of the contrasting estimates of MetS among the ethnicities, we explored the effect of ethnicity further with sociodemographic, lifestyle and diet factors. Table 3 shows the adjusted logistic regression models among the Malays, the Chinese and the Indians, revealing higher odds for MetS for quick finishing of meals among the Malays (AOR = 2.17 [95% CI = 1.02–4.60]) and low physical activity among the Chinese (AOR = 4.76 [95% CI = 1.49–15.26]). Furthermore, higher educational categories were protective against MetS among the Malays. Among the Indians, older age groups (40 years and above) were more prone to developing MetS, while significant odds with respect to age were found among the Chinese older than 60 years of age (AOR = 5.43 [95% CI = 1.39–21.13]).

## Discussion

In this study, the prevalence of MetS was found to be 32.2%, which was unexpectedly less than that reported in the 2008 nationwide survey, that contained 19% of subjects from Johor [11].

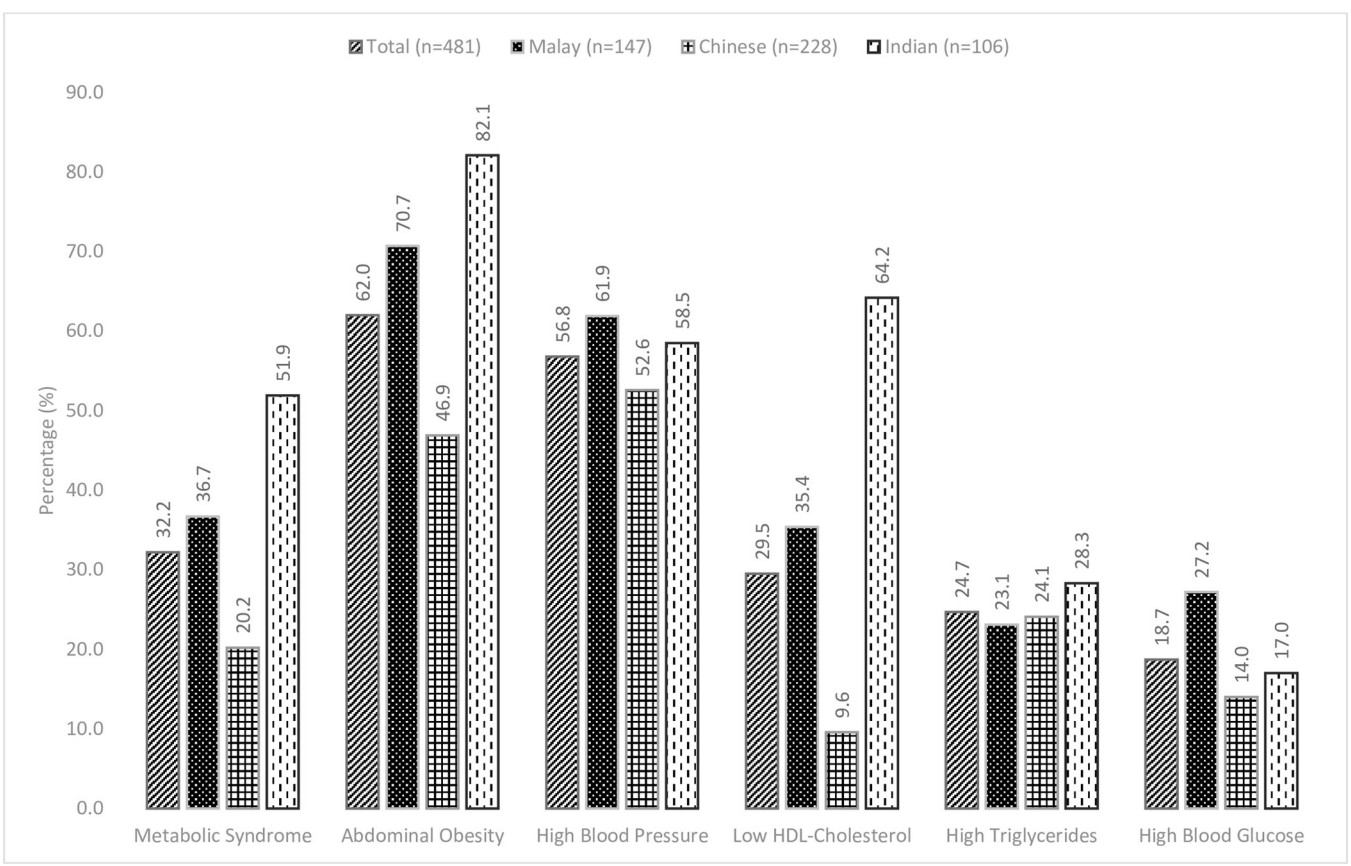

**Fig 1. Proportions of MetS and its components, overall and specifically among the three major ethnic groups in Johor.**

Prevalence among the Indians and the Chinese turned out to be 51.9% and 20.2%, respectively. Comparing the prevalence values reported in the nationwide survey 2008 for these two ethnic groups in Malaysia, the prevalence among the Indians appears to have remained unchanged over a period of nine years, while among the Chinese, the prevalence has reduced considerably from 42.1% to 20.2% [11]. However, among the Malays the prevalence has decreased from 43.9% to 36.7% [11]. This apparent decline among the Malays and the Chinese has been attributed to decreased prevalence values of hyperglycemia, low HDL-cholesterol and hypertriglyceridemia.

The present study showed that MetS was more prominent among the higher age groups. This finding has been observed by other researchers as well. He et al., reported a comparatively higher prevalence of MetS among older subjects (70 years and above) compared to those aged between 60–69 years among a total of over two thousand Chinese subjects. [37]. In the study by Rampal et al., the prevalence of MetS among the Malaysians was found to be higher among subjects aged 40 years and above compared to those aged less than 40 years (44.6% vs. 16.0%) [38]. Moreover, in the study by Ramli and colleagues, the odds of MetS, irrespective of definition applied, were found to be higher among higher age groups, and maximum among subjects aged 60 years and above [12]. The nationwide survey also reported higher prevalence of MetS among higher age groups; additionally, higher age groups also had a higher prevalence of central obesity, high blood pressure, low HDL-cholesterol, elevated triglycerides and hyperglycemia [11]. This suggests that higher prevalence of MetS among higher age groups may be due to the accumulated higher prevalence of its associated cardio-metabolic risk factors among elderly subjects.

Table 1. Summary of sociodemographic, lifestyle and dietary characteristics with MetS, overall and among the three major ethnic groups in Johor.

| | | Overall | | | Malay | | | Chinese | | | Indian | | |
|---|---|---|---|---|---|---|---|---|---|---|---|---|---|
| | | Total (n = 481) | With MetS (n = 155) | P-value* | Total (n = 147) | With MetS (n = 54) | P-value* | Total (n = 228) | With MetS (n = 46) | P-value* | Total (n = 106) | With MetS (n = 55) | P-value* |
| Characteristic | | n (%) | n (%) | | n (%) | n (%) | | n (%) | n (%) | | n (%) | n (%) | |
| Gender | Male | 169 (35.1) | 57 (36.8) | 0.604 | 42 (28.6) | 12 (22.2) | 0.194 | 88 (38.6) | 20 (43.5) | 0.447 | 39 (36.8) | 25 (45.5) | 0.055 |
| | Female | 312 (64.9) | 98 (63.2) | | 105 (71.4) | 42 (77.8) | | 140 (61.4) | 26 (56.5) | | 67 (63.2) | 30 (54.5) | |
| Age (years) | < 40 | 112 (23.3) | 22 (14.2) | 0.005 | 38 (25.9) | 12 (22.2) | 0.442 | 40 (17.5) | 3 (6.5) | 0.016 | 34 (32.1) | 7 (12.7) | < 0.001 |
| | 40–49 | 129 (26.8) | 48 (31.0) | | 56 (38.1) | 18 (33.3) | | 35 (15.4) | 7 (15.2) | | 38 (35.8) | 23 (41.8) | |
| | 50–59 | 117 (24.3) | 36 (23.2) | | 32 (21.8) | 14 (25.9) | | 71 (31.1) | 11 (23.9) | | 14 (13.2) | 11 (20.0) | |
| | ≥ 60 | 123 (25.6) | 49 (31.6) | | 21 (14.3) | 10 (18.5) | | 82 (36.0) | 25 (54.3) | | 20 (18.9) | 14 (25.5) | |
| Ethnicity | Malay | 147 (30.6) | 54 (34.8) | < 0.001 | - | - | - | - | - | - | - | - | - |
| | Chinese | 228 (47.4) | 46 (29.7) | | - | - | | - | - | | - | - | |
| | Indian | 106 (22.0) | 55 (35.5) | | - | - | | - | - | | - | - | |
| Marital status | Single | 63 (13.1) | 13 (8.4) | 0.002 | 5 (3.4) | 3 (5.6) | 0.035 | 49 (21.5) | 7 (15.2) | 0.284 | 9 (8.5) | 3 (5.5) | 0.420 |
| | Married/living with partner | 378 (78.6) | 123 (79.4) | | 126 (85.7) | 41 (75.9) | | 168 (73.7) | 38 (82.6) | | 84 (79.2) | 44 (80.0) | |
| | Separated/ divorced | 40 (8.3) | 19 (12.3) | | 16 (10.9) | 10 (18.5) | | 11 (4.8) | 1 (2.2) | | 13 (12.3) | 8 (14.5) | |
| Education | Primary or lower | 95 (19.8) | 47 (30.3) | < 0.001 | 32 (21.8) | 19 (35.2) | 0.004 | 35 (15.4) | 12 (26.1) | 0.062 | 28 (26.4) | 16 (29.1) | 0.255 |
| | Secondary | 291 (60.5) | 92 (59.4) | | 100 (68.0) | 33 (61.1) | | 126 (55.3) | 24 (52.2) | | 65 (61.3) | 35 (63.6) | |
| | Tertiary | 95 (19.8) | 16 (10.3) | | 15 (10.2) | 2 (3.7) | | 67 (29.4) | 10 (21.7) | | 13 (12.3) | 4 (7.3) | |
| Employment status | Employed | 206 (42.8) | 61 (39.4) | 0.289 | 47 (32.0) | 14 (25.9) | 0.231 | 111 (48.7) | 20 (43.5) | 0.429 | 48 (45.3) | 27 (49.1) | 0.413 |
| | Unemployed | 275 (57.2) | 94 (60.6) | | 100 (68.0) | 40 (74.1) | | 117 (51.3) | 26 (56.5) | | 58 (54.7) | 28 (50.9) | |
| Level of Physical activity | High | 95 (19.8) | 26 (16.8) | 0.045 | 66 (44.9) | 27 (50.0) | 0.520 | 81 (35.5) | 23 (50.0) | 0.029 | 47 (44.3) | 25 (45.5) | 0.957 |
| | Moderate | 192 (39.9) | 54 (34.8) | | 52 (35.4) | 16 (29.6) | | 102 (44.7) | 19 (41.3) | | 38 (35.8) | 19 (34.5) | |
| | Low | 194 (40.3) | 75 (48.4) | | 29 (19.7) | 11 (20.4) | | 45 (19.7) | 4 (8.7) | | 21 (19.8) | 11 (20.0) | |
| Smoking status | Non smoker | 419 (87.1) | 138 (89.0) | 0.386 | 124 (84.4) | 49 (90.7) | 0.104 | 205 (89.9) | 43 (93.5) | 0.369 | 90 (84.9) | 46 (83.6) | 0.705 |
| | Past/Current smoker | 62 (12.9) | 17 (11.0) | | 23 (15.6) | 5 (9.3) | | 23 (10.1) | 3 (6.5) | | 16 (15.1) | 9 (16.4) | |
| Alcohol consumption | Never/past consumer | 440 (91.5) | 139 (89.7) | 0.330 | 146 (99.3) | 49 (90.7) | 0.104 | 205 (89.9) | 43 (93.5) | 0.369 | 90 (84.9) | 46 (83.6) | 0.705 |
| | Current consumer | 41 (8.5) | 16 (10.3) | | 1 (0.7) | 1 (1.9) | | 23 (10.1) | 3 (6.5) | | 16 (15.1) | 9 (16.4) | |
| Frequency of dining out | ≤ 3 times/ week | 327 (68.0) | 110 (71.0) | 0.333 | 132 (89.8) | 47 (87.0) | 0.400 | 110 (48.2) | 22 (47.8) | 0.949 | 85 (80.2) | 41 (74.5) | 0.130 |
| | > 3 times/ week | 154 (32.0) | 45 (29.0) | | 15 (10.2) | 7 (13.0) | | 118 (51.8) | 24 (52.2) | | 21 (19.8) | 14 (25.5) | |
| Frequency of Late dining | ≤ 3 times/ week | 428 (89.0) | 136 (87.7) | 0.549 | 129 (87.8) | 49 (90.7) | 0.400 | 208 (91.2) | 41 (89.1) | 0.574 | 91 (85.8) | 46 (83.6) | 0.497 |
| | > 3 times/ week | 53 (11.0) | 19 (12.3) | | 18 (12.2) | 5 (9.3) | | 20 (8.8) | 5 (10.9) | | 15 (14.2) | 9 (16.4) | |
| Frequency of Skipping breakfast | ≤ 3 times/ week | 364 (75.7) | 119 (76.8) | 0.699 | 117 (79.6) | 42 (77.8) | 0.678 | 175 (76.8) | 37 (80.4) | 0.508 | 72 (67.9) | 40 (72.7) | 0.271 |
| | > 3 times/ week | 117 (24.3) | 36 (23.2) | | 30 (20.4) | 12 (22.2) | | 53 (23.2) | 9 (19.6) | | 34 (32.1) | 15 (27.3) | |

(*Continued*)

**Table 1.** (Continued)

| | | Overall | | | Malay | | | Chinese | | | Indian | | |
|---|---|---|---|---|---|---|---|---|---|---|---|---|---|
| | | Total (n = 481) | With MetS (n = 155) | P-value* | Total (n = 147) | With MetS (n = 54) | P-value* | Total (n = 228) | With MetS (n = 46) | P-value* | Total (n = 106) | With MetS (n = 55) | P-value* |
| Characteristic | | n (%) | n (%) | | n (%) | n (%) | | n (%) | n (%) | | n (%) | n (%) | |
| Rate of finishing meals | Not fast | 229 (47.6) | 66 (42.6) | 0.128 | 79 (53.7) | 24 (44.4) | 0.085 | 121 (53.1) | 24 (52.2) | 0.892 | 29 (27.4) | 18 (32.7) | 0.198 |
| | Fast | 252 (52.4) | 89 (57.4) | | 68 (46.3) | 30 (55.6) | | 107 (46.9) | 22 (47.8) | | 77 (72.6) | 37 (67.3) | |

*P-value ascertained by $X^2$ test.

**Table 2. Association of sociodemographic and lifestyle factors and diet habits with MetS among the overall population of Johor (n = 481).**

| Characteristic | | Crude OR (95% CI) | Adjusted OR (95% CI) |
|---|---|---|---|
| Gender | Male | 1.00 | |
| | Female | 0.90 (0.60–1.34) | |
| Age (years) | < 40 | 1.00 | 1.00 |
| | 40–49 | 2.42 (1.35–4.36)* | 2.44 (1.27–4.70)* |
| | 50–59 | 1.82 (0.99–3.34) | 2.80 (1.36–5.79)* |
| | ≥ 60 | 2.71 (1.50–4.88)* | 4.14 (1.97–8.69)* |
| Ethnicity | Malay | 1.00 | 1.00 |
| | Chinese | 0.44 (0.27–0.69)* | 0.37 (0.21–0.65)* |
| | Indian | 1.86 (1.12–3.08)* | 1.95 (1.12–3.38)* |
| Marital status | Single | 1.00 | 1.00 |
| | Married/living with partner | 1.86 (0.97–3.54) | 0.81 (0.38–1.71) |
| | Separated/divorced | 3.48 (1.46–8.31)* | 0.75 (0.26–2.19) |
| Education | Primary or lower | 1.00 | 1.00 |
| | Secondary | 0.47 (0.29–0.76)* | 0.63 (0.37–1.08) |
| | Tertiary | 0.21 (0.11–0.40)* | 0.44 (0.20–0.94)* |
| Employment status | Employed | 1.00 | |
| | Unemployed | 1.23 (0.84–1.82) | |
| Level of Physical activity | High | 1.00 | 1.00 |
| | Moderate | 1.67 (0.98–2.86) | 1.14 (0.63–2.04) |
| | Low | 1.04 (0.60–1.80) | 1.77 (0.99–3.16) |
| Smoking status | Non smoker | 1.00 | |
| | Past/Current smoker | 0.77 (0.42–1.39) | |
| Alcohol consumption | Never consumed/past consumer | 1.00 | |
| | Current consumer | 1.39 (0.72–2.68) | |
| Frequency of dining out | ≤ 3 times/week | 1.00 | |
| | > 3 times/week | 0.81 (0.54–1.24) | |
| Frequency of Late dining | ≤ 3 times/week | 1.00 | |
| | > 3 times/week | 1.20 (0.66–2.18) | |
| Frequency of Skipping breakfast | ≤ 3 times/week | 1.00 | |
| | > 3 times/week | 0.92 (0.58–1.43) | |
| Rate of finishing meals | Not fast | 1.00 | 1.00 |
| | Fast | 1.35 (0.92–1.98) | 1.15 (0.75–1.78) |

OR = Odds ratio; CI = Confidence interval; Empty cells refers to the variables excluded (p > 0.25 on univariate analyses) from the model.
*P-value < 0.05

**Table 3. Association of sociodemographic and lifestyle factors and diet habits with MetS among the three major ethnicities of Johor.**

| Characteristic | | Malay (n = 147) | | Chinese (n = 228) | | Indian (n = 106) | |
|---|---|---|---|---|---|---|---|
| | | Crude OR (95% CI) | Adjusted OR (95% CI) | Crude OR (95% CI) | Adjusted OR (95% CI) | Crude OR (95% CI) | Adjusted OR (95% CI) |
| Gender | Male | 1.00 | 1.00 | 1.00 | | 1.00 | 1.00 |
| | Female | 1.67 (0.77–3.62) | 1.16 (0.35–2.85) | 0.78 (0.40–1.49) | | 0.45 (0.20–1.02) | 0.62 (0.22–1.80) |
| Age (years) | < 40 | 1.00 | | 1.00 | 1.00 | 1.00 | 1.00 |
| | 40–49 | 1.03 (0.42–2.49) | | 3.08 (0.73–13.00) | 3.08 (0.69–13.62) | 5.91 (2.06–16.99)* | 8.24 (2.38–28.59)* |
| | 50–59 | 1.69 (0.63–4.48) | | 2.26 (0.59–8.64) | 2.43 (0.60–9.86) | 14.14 (3.08–64.88)* | 25.25 (4.64–137.57)* |
| | ≥ 60 | 1.97 (0.66–5.89) | | 5.41 (1.52–19.20)* | 5.43 (1.39–21.13)* | 9.00 (2.54–31.96)* | 14.25 (3.22–63.84)* |
| Marital status | Single | 1.00 | 1.00 | 1.00 | | 1.00 | |
| | Married/living with partner | 0.32 (0.05–2.00) | 0.16 (0.02–1.27) | 1.75 (0.73–4.22) | | 2.20 (0.52–9.38) | |
| | Separated/divorced | 1.11 (0.14–8.68) | 0.33 (0.03–3.63) | 0.60 (0.07–5.45) | | 3.20 (0.54–18.98) | |
| Education | Primary or lower | 1.00 | 1.00 | 1.00 | 1.00 | 1.00 | |
| | Secondary | 0.34 (0.15–0.77)* | 0.38 (0.15–0.95)* | 0.45 (0.20–1.03) | 0.64 (0.26–1.58) | 0.88 (0.36–2.14) | |
| | Tertiary | 0.10 (0.02–0.55)* | 0.08 (0.01–0.58)* | 0.34 (0.13–0.87)* | 0.54 (0.19–1.57) | 0.33 (0.08–1.35) | |
| Employment status | Employed | 1.00 | 1.00 | 1.00 | | 1.00 | |
| | Unemployed | 1.57 (0.75–3.30) | 0.78 (0.30–2.09) | 1.30 (0.68–2.49) | | 0.73 (0.34–1.56) | |
| Level of Physical activity | High | 1.00 | | 1.00 | 1.00 | 1.00 | |
| | Moderate | 0.73 (0.28–1.89) | | 2.35 (0.75–7.35) | 2.30 (0.72–7.34) | 0.91 (0.31–2.64) | |
| | Low | 1.13 (0.46–2.78) | | 4.06 (1.31–12.64)* | 4.76 (1.49–15.26)* | 1.03 (0.37–2.89) | |
| Smoking status | Non Smoker | 1.00 | 1.00 | 1.00 | | 1.00 | |
| | Past/Current smoker | 0.42 (0.15–1.22) | 0.60 (0.15–2.42) | 0.56 (0.16–1.99) | | 1.23 (0.42–3.58) | |
| Alcohol consumption | Never/past consumer | 1.00 | | 1.00 | | 1.00 | |
| | Current consumer | - | | 0.99 (0.38–2.58) | | 9.78 (1.19–80.24)* | |
| Frequency of dining out | ≤ 3 times/week | 1.00 | | 1.00 | | 1.00 | 1.00 |
| | > 3 times/week | 1.58 (0.54–4.64) | | 1.02 (0.54–1.95) | | 2.15 (0.79–5.85) | 4.18 (0.98–17.83) |
| Frequency of Late dining | ≤ 3 times/week | 1.00 | | 1.00 | | 1.00 | |
| | > 3 times/week | 0.63 (0.21–1.87) | | 1.36 (0.47–3.95) | | 1.47 (0.48–4.46) | |
| Frequency of Skipping breakfast | ≤ 3 times/week | 1.00 | | 1.00 | | 1.00 | |
| | > 3 times/week | 1.19 (0.52–2.71) | | 0.76 (0.34–1.70) | | 0.63 (0.28–1.44) | |
| Rate of finishing meals | Not fast | 1.00 | 1.00 | 1.00 | | 1.00 | 1.00 |
| | Fast | 1.81 (0.92–3.56) | 2.17 (1.02–4.60)* | 1.05 (0.55–1.99) | | 0.56 (0.24–1.35) | 0.68 (0.25–1.81) |

OR = Odds ratio; CI = Confidence interval; Empty cells refers to the variables excluded (p > 0.25 on univariate analyses) from the model.

*P-value < 0.05

Studies have shown that ethnicity influences the prevalence of metabolic syndrome. For instance, from a study in Canada, MetS prevalence was reported to be higher among the Cree Indians compared to other aboriginal and non-aboriginal Canadians [14]. The Cree Indians also had a higher prevalence of central obesity and hyperglycemia compared to other ethnic groups in the country [14]. Similarly, from a study in Suriname, South America, MetS prevalence was reported as the highest among the Hindustanis (descendent of Indians), compared to other Suriname races [39]. The prevalence values of high blood pressure, low HDL-cholesterol and hyperglycemia were also high among the Suriname Hindustanis [39]. In a local study among obese adolescents, Indians again had the highest prevalence of MetS, contributed mainly by higher prevalence of central obesity, increased blood pressure and low HDL-cholesterol [40]. In our study, results show that the Indians in Johor are at a greater risk of developing MetS, while the Chinese appear to be less prone to developing MetS. This is in line with the reports from other researchers from Malaysia that the Chinese have lower odds, while the Indians have higher odds of developing MetS [12, 38]. More educated adults in the Johor area, especially the Malays, appear to be protected against MetS, probably due to their increased awareness of healthy lifestyle habits, such as engagement in physical activity, smoking cessation, moderate to none consumption of alcohol and adoption of healthy eating habits [26]. This is supported by a couple of studies showing that the individual's higher level of education is protective against diabetes and hypertension, which are prominent risk components of MetS [41, 42]. Furthermore Kaur et al., reported that the odds of MetS among Malaysian Punjabis (Indians) were higher with primary education, compared to those with higher education levels [43]. Ching et al., however, have reported that higher education levels of Malaysian vegetarians with and without MetS were observed to be nearly the same [44]. This could be due to the fact that it was a unique group of subjects with specific dietary habits and the results pertaining to this group may not represent the general population of Malaysia.

Literature suggests that excess energy accumulated in the adipose tissues causes metabolic abnormalities, leading to high blood pressure, hyperglycemia, hypertriglyceridemia and inflammation, hence, regular physical activity enhances energy consumption leading to reduced prevalence of obesity, hypertension, diabetes mellitus and also MetS [45, 46]. Our results show the prevalence of MetS and its components to be comparatively lower among the Chinese than in the Malays and the Indians, and this could be attributed to better physical activity among them. Chu et al., have shown that longer sitting time and insufficient physical activity have resulted in an almost 4-fold increase in MetS risk among the Malays, and the risk gets reduced by 50% by engaging in moderate to high physical activity [27, 28]. On the basis of these reports, it can be suggested that despite having a decreased risk of developing MetS, the Chinese in Johor can still benefit by engaging in moderate to high levels of physical activity.

A number of studies have shown a direct relationship of smoking with the risk of MetS, yet in the current study smoking does not appear to be associated with the risk of MetS [20]. This could be due to a small percentage of past and current smokers (12.9%) in this cohort. Similarly, no association was found between alcohol consumption and risk of MetS among those in this population. Again, the reason could be the small proportion of subjects who were reported as alcohol consumers (8.5%).

The association of dietary habits, such as quick finishing of meals, frequent dining out, late eating, skipping breakfast, with MetS has been reported in other studies in the East Asian region [25, 26, 47–50]. For example, Shin et al., reported quick eating as one of the risk factors for MetS among the Koreans [50]. Among these dietary habits, quick finishing of meals was identified in the current study as a new risk factor for MetS in Malaysia, especially among the Malays. According to Dallman and colleagues, fast eaters may consume more food than usual, or be eating under psychological stress which affects hormones regulating metabolism [51].

The underlying mechanism of relationship of such habit(s) with the metabolic health functioning, however remains unclear [50].

There are certain limitations that warrant consideration. First, the present research study was cross-sectional in nature, assessing the exposures and outcomes at the same point in time. In this regard, the findings cannot indicate causality. Second, the selection of study locations harboring subjects was non-random and partly based on the information on the available percentage of MetS across each Malaysian ethnicity in Johor as reported in the nationwide survey 2008 [11]. This was done to have sufficient number of subjects in each ethnic group for better analysis and interpretation, especially among the minority ethnic groups, which had been underrepresented in previous research from Malaysia. Though ethnic stratification was never intended, the in-depth analysis showed the variable influences of lifestyle and diet habits on MetS, especially among the Indians and the Chinese with adequate power, suggesting that the variable effects were more likely attributed to the cultural diversities across the different ethnicities of Malaysia. Moreover, as the information collected was based on recall, misreporting of information cannot be completely ruled out, and this might have added some variability in our results. Despite these sources of potential variability, the results provide credible evidence towards the association of certain sociodemographic, lifestyle and diet factors that affect the disease spectrum of MetS in Johor, and provided an opportunity to further analyze these characteristics influencing MetS across the three major ethnic groups of this state in the country. In this regard, we believe the current research study to be adequate and its findings comparable to similar studies by other investigators using a non-randomized design and exploring associations of various risk factors influencing metabolic diseases in Malaysia.

Based on the results of this study, it can be deduced that the population of Johor is diverse in its habits pertaining to lifestyle and diet. Some of these factors are associated with the risk of MetS in certain ethnic groups and modifying these factors would be important for reducing cardiovascular and metabolic health risks among Malaysians. Though not analyzed in detail in this research, effects of other variables determining socioeconomic position, like wealth index, could be explored with MetS in future research. Moreover, further prospective studies delineating the association of various diet habits among different ethnic communities would be imperative to contain the unfavorable effects of this syndrome on the overall health of Malaysians. Since Malaysia is a multi-ethnic country, it would be important to consider this ethnic variation, especially with respect to lifestyle and diet factors, so that intervention programs for addressing behavior modifications would also be tailored across different ethnicities of the country. Increasing awareness among the masses through electronic and print media about the beneficial effects of healthy lifestyle is likely to be another powerful approach to combat the menace of this syndrome in Malaysia.

## Supporting information

**S1 File. Questionnaire.**
(DOCX)

**S2 File. Dataset.**
(XLSX)

## Acknowledgments

The authors wish to thank Gribbles Laboratories (MS ISO 15189), Malaysia, for assisting with the laboratory assessments on the collected blood samples. We would also like to extend our gratitude to Government contacts (Majlis Perbandaran Kulai), local community leaders and

the Monash University faculty and staff, namely Mr Chui Chor Sin, Mrs Savithri Gopal, Ms Pang Pei Ling, Ms Ungku Zulaikha Ungku Omar, Mr Muhammad Daniel Mahadzir, Dr Nor Azim, Ms Kong Li San and Dr Iekhsan Othman for all their support and assistance in this study. Additionally, the authors would also like to thank Ms Harbans Kaur Singara Singh (Education Development Division, LeapEd Services Sdn. Bhd.) for proofreading the manuscript.

## Author Contributions

**Conceptualization:** Saleem Perwaiz Iqbal, Amutha Ramadas, Quek Kia Fatt, Khalid Abdul Kadir.

**Data curation:** Saleem Perwaiz Iqbal.

**Formal analysis:** Saleem Perwaiz Iqbal, Khalid Abdul Kadir.

**Funding acquisition:** Amutha Ramadas, Khalid Abdul Kadir.

**Investigation:** Saleem Perwaiz Iqbal, Ho Loon Shin, Wong Yin Onn, Khalid Abdul Kadir.

**Methodology:** Saleem Perwaiz Iqbal, Amutha Ramadas, Ho Loon Shin, Wong Yin Onn, Khalid Abdul Kadir.

**Project administration:** Saleem Perwaiz Iqbal, Amutha Ramadas, Ho Loon Shin, Wong Yin Onn, Khalid Abdul Kadir.

**Resources:** Ho Loon Shin, Wong Yin Onn, Khalid Abdul Kadir.

**Supervision:** Amutha Ramadas, Quek Kia Fatt, Khalid Abdul Kadir.

**Writing – original draft:** Saleem Perwaiz Iqbal.

**Writing – review & editing:** Saleem Perwaiz Iqbal, Amutha Ramadas, Quek Kia Fatt, Khalid Abdul Kadir.

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
