## [Decision Letter · Decision Letter 0]

11 Nov 2019

PONE-D-19-27618

Relationship of sociodemographic and lifestyle factors and diet habits with metabolic syndrome (MetS) in a multi-ethnic Asian population

PLOS ONE

Dear Dr Iqbal,

Thank you for submitting your manuscript to PLOS ONE. After careful consideration, we feel that it has merit but does not fully meet PLOS ONE’s publication criteria as it currently stands. Therefore, we invite you to submit a revised version of the manuscript that addresses the points raised during the review process.

We would appreciate receiving your revised manuscript by Dec 26 2019 11:59PM. To enhance the reproducibility of your results, we recommend that if applicable you deposit your laboratory protocols in protocols.io, where a protocol can be assigned its own identifier (DOI) such that it can be cited independently in the future. For instructions see: http://journals.plos.org/plosone/s/submission-guidelines#loc-laboratory-protocols

We look forward to receiving your revised manuscript.

Kind regards,

Hoh Boon-Peng, PhD

Academic Editor

PLOS ONE

Journal Requirements:

3. In your Methods section, please provide additional information about the participant recruitment method and the demographic details of your participants. Please ensure you have provided sufficient details to replicate the analyses such as: a) the recruitment date range (month and year), b) a description of any inclusion/exclusion criteria that were applied to participant recruitment, c) a table of relevant demographic details, d) a statement as to whether your sample can be considered representative of a larger population, e) a description of how participants were recruited, and f) descriptions of where participants were recruited and where the research took place.

4. Please include additional information regarding the survey or questionnaire used in the study and ensure that you have provided sufficient details that others could replicate the analyses. For instance, if you developed a questionnaire as part of this study and it is not under a copyright more restrictive than CC-BY, please include a copy, in both the original language and English, as Supporting Information. Moreover, please include more details on how the questionnaire was pre-tested, and whether it was validated.

5. Please note that according to our submission guidelines (http://journals.plos.org/plosone/s/submission-guidelines), outmoded terms and potentially stigmatizing labels should be changed to more current, acceptable terminology. For example: “Caucasian” should be changed to “white” or “of [Western] European descent” (as appropriate).

6. Our internal editors have looked over your manuscript and determined that it is within the scope of our Health Inequities and Disparities Research Call for Papers. This collection of papers is headed by a team of Guest Editors for PLOS ONE: Clare Bambra, Hans Bosma, Diana Burgess, Joseph Telfair, Barbara Turner, and Jennie Popay. The Collection will encompass a diverse range of research articles on health inequities and disparities.  Additional information can be found on our announcement page: https://collections.plos.org/s/health-inequities.

If you would like your manuscript to be considered for this collection, please let us know in your cover letter and we will ensure that your paper is treated as if you were responding to this call. If you would prefer to remove your manuscript from collection consideration, please specify this in the cover letter.

7. Thank you for including your ethics statement:  "Monash University Human Research Ethics Committee (Project # CF15/56-2016000022). All subjects provided written consent for their participation in the study."

a) Please amend your current ethics statement to confirm that your named institutional review board or ethics committee specifically approved this study.

Additional Editor Comments:

Authors should emphasise the implications based on the current findings, and the future direction of the study.

Reviewers' comments:

Reviewer's Responses to Questions

**Comments to the Author**

1. Is the manuscript technically sound, and do the data support the conclusions?

Reviewer #1: Yes

Reviewer #2: Partly

2. Has the statistical analysis been performed appropriately and rigorously? 

Reviewer #1: I Don't Know

Reviewer #2: Yes

3. Have the authors made all data underlying the findings in their manuscript fully available?

Reviewer #1: Yes

Reviewer #2: Yes

4. Is the manuscript presented in an intelligible fashion and written in standard English?

Reviewer #1: Yes

Reviewer #2: No

5. Review Comments to the Author

Reviewer #1: The paper is well written highlighting the relevance of the study. The results were discussed in well and in detail. A minor suggestion would be to address a few grammatical errors and to suggest future research direction.

Reviewer #2: By using statistical test and multiple logistic regression, authors studied the association of multiple factors including sociodemographics, lifestyle, diet patterns with MetS of major ethnic populations in Malaysia. What is the potential clinical value of the finding to society in Malaysia?

Multiple grammatical errors were spotted and sentences are incoherent, it is strongly recommended to proof read the article by native speaker. Also, authors are advised to present the paper in reader friendly manner, for example tab indent at the beginning of new paragraph or newline insert between paragraphs.

Unclear conclusion with hanging sentences.

6. PLOS authors have the option to publish the peer review history of their article (what does this mean?). If published, this will include your full peer review and any attached files.

Reviewer #1: No

Reviewer #2: No

---

## [Author Response · Author response to Decision Letter 0]

22 Nov 2019

Answer: We thank the editor for pointing us this oversight. The revised manuscript is now according to the PLoS One guidelines. Please see the revised manuscript.

Answer: The captions have been added in the revised manuscript. Please see page 30 of the revised manuscript.

3. In your Methods section, please provide additional information about the participant recruitment method and the demographic details of your participants. Please ensure you have provided sufficient details to replicate the analyses such as: a) the recruitment date range (month and year), b) a description of any inclusion/exclusion criteria that were applied to participant recruitment, c) a table of relevant demographic details, d) a statement as to whether your sample can be considered representative of a larger population, e) a description of how participants were recruited, and f) descriptions of where participants were recruited and where the research took place.

Answer: The revised manuscript now contains additional details inclusive of:

1) recruitment date range (Please see page 8, under the sub-heading “Study design and location”, lines 2-4.).

2) description of inclusion/exclusion criteria (Please see page 8, under the subheading “Recruitment and eligibility criteria”, 2nd paragraph.).

3) description of relevant demographics (table 1 contains the demographic details) (Please see page 14, under the heading of “Results”, 2nd paragraph.).

4) Statement that the sample represents the Johor population, especially with respect to ethnicities (Please see the page 8, under the sub-heading “Study design and location”, lines 7-9.).

5) descriptions of how and where the participants were recruited and where the research took place (Please see the highlighted lines on page 8-9 of the revised manuscript with highlighted changes.).

4. We suggest you thoroughly copyedit your manuscript for language usage, spelling, and grammar. If you do not know anyone who can help you do this, you may wish to consider employing a professional scientific editing service. 

Answer: We thank the editor for this suggestion. This has been done as well in the revised manuscript. The copyediting of the manuscript was requested from: 

Ms Harbans Kaur Singara Singh

Email: harbans.singh@leapedservices.com

Please see the supporting file 3 (S3 File. Proofreading of manuscript with changes) and the revised manuscript.

5. Please include additional information regarding the survey or questionnaire used in the study and ensure that you have provided sufficient details that others could replicate the analyses. For instance, if you developed a questionnaire as part of this study and it is not under a copyright more restrictive than CC-BY, please include a copy, in both the original language and English, as Supporting Information. Moreover, please include more details on how the questionnaire was pre-tested, and whether it was validated.

Answer: We have included the bilingual version of questionnaire as a supporting file. We have also included details of pretest that was done before the actual data collection (Please see supporting file 1 (S1 File. Questionnaire). 

For details on pretesting, please see page 9 of the revised manuscript with highlighted changes, under the sub-heading “Data collection and measurement”, 1st paragraph, lines 5-7.

6. Please note that according to our submission guidelines (http://journals.plos.org/plosone/s/submission-guidelines), outmoded terms and potentially stigmatizing labels should be changed to more current, acceptable terminology. For example: “Caucasian” should be changed to “white” or “of [Western] European descent” (as appropriate).

Answer: We thank the editors and reviewers for this oversight. We have modified our sentences accordingly. (Please see the revised manuscript.)

7. Our internal editors have looked over your manuscript and determined that it is within the scope of our Health Inequities and Disparities Research Call for Papers. This collection of papers is headed by a team of Guest Editors for PLOS ONE: Clare Bambra, Hans Bosma, Diana Burgess, Joseph Telfair, Barbara Turner, and Jennie Popay. The Collection will encompass a diverse range of research articles on health inequities and disparities. Additional information can be found on our announcement page: https://collections.plos.org/s/health-inequities.

If you would like your manuscript to be considered for this collection, please let us know in your cover letter and we will ensure that your paper is treated as if you were responding to this call. If you would prefer to remove your manuscript from collection consideration, please specify this in the cover letter.

Answer: We thank PLoS One in this regard. We will certainly like our manuscript to be considered for this collection. We have specified this in the new cover letter. (Please see the new cover letter [Cover letter 2]).

8. Thank you for including your ethics statement: "Monash University Human Research Ethics Committee (Project # CF15/56-2016000022). All subjects provided written consent for their participation in the study."

a) Please amend your current ethics statement to confirm that your named institutional review board or ethics committee specifically approved this study.

Answer: This revision has been made. Please see page 13 of the revised manuscript; under the sub-heading “Ethics”, lines 1-2. Please also see the ethics statement of the new submission form.

9. (Reviewer 1) The paper is well written highlighting the relevance of the study. The results were discussed in well and in detail. A minor suggestion would be to address a few grammatical errors and to suggest future research direction.

Answer: Grammatical errors have been addressed and future research direction emphasised (Please see page 26 of the revised manuscript.).

10. (Reviewer 2) By using statistical test and multiple logistic regression, authors studied the association of multiple factors including sociodemographics, lifestyle, diet patterns with MetS of major ethnic populations in Malaysia. What is the potential clinical value of the finding to society in Malaysia? Multiple grammatical errors were spotted and sentences are incoherent, it is strongly recommended to proof read the article by native speaker. Also, authors are advised to present the paper in reader friendly manner, for example tab indent at the beginning of new paragraph or newline insert between paragraphs. Unclear conclusion with hanging sentences.

Answer: We have added sentences to answer the question the reviewer has asked (Please see page 26, lines 1-7 of the revised manuscript with highlighted changes.). Furthermore, Grammatical errors have been addressed, presentation of the paper has been modified and conclusions has been revised (Please see page 26, lines 1-7 of the revised manuscript with highlighted changes).

---

## [Decision Letter · Decision Letter 1]

10 Jan 2020

PONE-D-19-27618R1

Relationship of sociodemographic and lifestyle factors and diet habits with metabolic syndrome (MetS) in a multi-ethnic Asian population

PLOS ONE

Dear Dr Iqbal,

Thank you for submitting your manuscript to PLOS ONE. After careful consideration, we feel that it has merit but does not fully meet PLOS ONE’s publication criteria as it currently stands. Therefore, we invite you to submit a revised version of the manuscript that addresses the points raised during the review process.

We would appreciate receiving your revised manuscript by Feb 24 2020 11:59PM. To enhance the reproducibility of your results, we recommend that if applicable you deposit your laboratory protocols in protocols.io, where a protocol can be assigned its own identifier (DOI) such that it can be cited independently in the future. For instructions see: http://journals.plos.org/plosone/s/submission-guidelines#loc-laboratory-protocols

We look forward to receiving your revised manuscript.

Kind regards,

Hoh Boon-Peng, PhD

Academic Editor

PLOS ONE

Additional Editor Comments (if provided):

Although sample size calculation was included, the issue on small sample size of this study remains a question. Authors could justify by showing the prevalence of MetS for each ethnicities, and the calculation of power of studies.

Reviewers' comments:

Reviewer's Responses to Questions

**Comments to the Author**

1. If the authors have adequately addressed your comments raised in a previous round of review and you feel that this manuscript is now acceptable for publication, you may indicate that here to bypass the “Comments to the Author” section, enter your conflict of interest statement in the “Confidential to Editor” section, and submit your "Accept" recommendation.

Reviewer #3: (No Response)

Reviewer #4: All comments have been addressed

2. Is the manuscript technically sound, and do the data support the conclusions?

Reviewer #3: Partly

Reviewer #4: Yes

3. Has the statistical analysis been performed appropriately and rigorously? 

Reviewer #3: Yes

Reviewer #4: Yes

4. Have the authors made all data underlying the findings in their manuscript fully available?

Reviewer #3: Yes

Reviewer #4: Yes

5. Is the manuscript presented in an intelligible fashion and written in standard English?

Reviewer #3: Yes

Reviewer #4: Yes

6. Review Comments to the Author

Reviewer #3: The paper described a cross-sectional study of 481 participants on the association between metabolic syndrome and sociodemographic, lifestyle factors and diet habits. The authors collected data via questionnaire survey at 5 study sites in the state of Johor from 2016 to 2018. Based on the findings, the authors have concluded that three Malaysian ethnic groups are diverse in their sociodemographic, lifestyle factors and diet habits and therefore showed difference in the prevalence of MetS. In general, older age is highly associated with the prevalence of MetS; MetS is more and less prevalent to Indian and Chinese, respectively; higher education decreased the risk of developing MetS; and quick finishing of meals and low physical activity is associated with the development of MetS.

However, there is a major concern regarding the sample size and the interpretation of data in the paper. The total number of samples is 481 and when it was stratified into 3 ethnic groups, Malay, Chinese and Indian have the number of 147, 228 and 106, respectively. These sample sizes are too small and cannot be settled with valid and reliable results, especially for Table 3. The problem of small sample size is simply reflected on the wide 95% CI in Table 3 where the authors claimed that the higher age groups of Chinese and Indian were associated with MetS, low physical activity in Chinese and fast finishing meals in Malay were in favour of greater risk of MetS. As the 95% CI is extremely wide, it indicates the very weak association of variables with MetS despite the p value is less than 0.05. In addition, the method in calculating the sample size was not written in detail and also it is not clear how the estimated sample size was calculated for each ethnic group.

Another concern would be the variables studied by the authors. A few relevant variables are missing in the analysis. Among them, wealth/income index is the most relevant variable for the sociodemographic characteristics but it is not included in the analysis.

Some minor comments:

(1) The title of paper does not reflect the actual population that was studied. I would suggest to replace “a multi-ethnic Asian population” to “three ethnic groups of the Malaysian population” in the title.

(2) The study was taken place in Johor, however, no explanation of why Johor is selected in this study. Can the population in Johor be the representative of the nation?

(3) page 15, line 3 from the bottom of paragraph: “….41.6% at the age group of 40-49 years…..” Instead of 41.6%, it was 41.8% written in Table 1.

(4) page 24, line 4 from the top of paragraph: “Our results show the prevalence of MetS and its components to be comparatively lower among Chinese than in the Malays and the Indians, and this could be attributed to better lifestyle choices, including physical activity”. The sentence is overstated since there is not enough evidence to show Chinese have better lifestyle choices except the physical activity.

(5) The following references are similar studies and shall be mentioned in the paper:

- Kaur et al. (2018) Journal of Immigrant and Minority Health 20, 1380-1386.

- Narayanan et al. (2011) Metabolic Syndrome and Related Disorders 9, 389-395.

- Tan et al. (2011) Metabolic Syndrome and Related Disorders 9, 441-451.

Reviewer #4: The paper is well written; the significance of the study is clearly indicated. The results and discussion are adequately presented and described. However, some defining criteria of metabolic syndrome (page 12) are inaccurately written - the definitions of raised serum triglycerides and low HDL are missing the symbols of "equals to or more than"and "less than" respectively. The grammatical deficiency has been addressed immensely. However, isolated grammatical errors are still present including those listed below:

1)page 23; paragraph 2; line 1

2)page 24; paragraph 1; line 10

2) page 25; paragraph 2; line 6-7

7. PLOS authors have the option to publish the peer review history of their article (what does this mean?). If published, this will include your full peer review and any attached files.

Reviewer #3: No

Reviewer #4: No

---

## [Author Response · Author response to Decision Letter 1]

20 Feb 2020

We thank the editor and reviewers for their kind feedback on improving the manuscript. Our reply to the concerns mentioned are as below:

1. Although sample size calculation was included, the issue on small sample size of this study remains a question. Authors could justify by showing the prevalence of MetS for each ethnicities, and the calculation of power of studies.

Answer: Prevalence of MetS across ethnicities have been added in the Results section. 

While calculating the sample size at the start of the study, stratification was not the objective of the study. 

However, post study calculation for Power has been performed for the 3 major ethnic groups in Johor revealing that except for the Malays, the study power for the Indians and the Chinese was pretty good (84.8% and 71.7%, respectively). In the previous local research on MetS in Johor, these minor ethnic groups had been underrepresented, however, in this study we ensured adequate representation of these minor ethnicities, so that associations could be seen also among all 3 major races residing in Johor. We have indicated this in the discussion section of the revised manuscript.

Please see the highlighted lines (line 8-10) of the last paragraph on page 25 of the revised manuscript.

2. There is a major concern regarding the sample size and the interpretation of data in the paper. The total number of samples is 481 and when it was stratified into 3 ethnic groups, Malay, Chinese and Indian have the number of 147, 228 and 106, respectively. These sample sizes are too small and cannot be settled with valid and reliable results, especially for Table 3. The problem of small sample size is simply reflected on the wide 95% CI in Table 3 where the authors claimed that the higher age groups of Chinese and Indian were associated with MetS, low physical activity in Chinese and fast finishing meals in Malay were in favour of greater risk of MetS. As the 95% CI is extremely wide, it indicates the very weak association of variables with MetS despite the p value is less than 0.05. In addition, the method in calculating the sample size was not written in detail and also it is not clear how the estimated sample size was calculated for each ethnic group.

Answer: Most respectfully, we would beg to disagree with the reviewer on this point. The results showed in table 3 needs to be seen in the continuation of table 2. Table 2 shows the effect of lifestyle and diet factors with MetS overall. For example, overall, increasing age increases the odds of MetS (a significant finding with precise confidence intervals). This characteristic was then further viewed within each ethnicity, yielding results that corroborated with the findings in table 2.

Stratification was never intended for this study The objective was to see which ethnicity or ethnicities influence the outcome in presence of the risk factor, as seen in the overall target sample. However, we do agree that stratification does lower the study power and widens the confidence intervals indicating weak associations (as shown among Indians with age in our study). Despite the weaker association, the study does indicate that an association exists. Other researchers have also shown these findings in research conducted in similar settings. 

We have modified our sentences in the discussion section to highlight this aspect and indicated that for future research on such populations stratification by ethnicity is likely to yield more conclusive results. Please see the highlighted lines (line 6-12) of the last paragraph of the Discussion section on page 26 in the revised manuscript. 

3. A few relevant variables are missing in the analysis. Among them, wealth/income index is the most relevant variable for the sociodemographic characteristics but it is not included in the analysis.

Answer: We agree that the wealth index is important to explore the sociodemographic characteristics. However, this is a composite variable; requiring detailed information on the households’ assets, house construction, water access, etc, which was beyond the scope of this research.

We had a question in our questionnaire to inquire the monthly incomes. However, this being a sensitive question, more than half of the recruited participants refused to furnish this information. Consequently, this variable could not be included in the final analysis. 

We have indicated in the discussion section that the association of wealth index with MetS needs to be explored in future research. Please see lines 4-6 of the last paragraph of the Discussion section on page 26 of the revised manuscript.

4. The title of paper does not reflect the actual population that was studied. I would suggest to replace “a multi-ethnic Asian population” to “three ethnic groups of the Malaysian population” in the title.

Answer: We greatly appreciate this useful suggestion and have made this modification, and the title has been changed as suggested.Please see the highlighted title on page 1 of the revised manuscript.

5. The study was taken place in Johor, however, no explanation of why Johor is selected in this study. Can the population in Johor be the representative of the nation?

Answer: Johor represents the other states in Malaysia in terms of ethnic distribution, based on statistics by the Department of Statistics, Malaysia.

As we employed a non-random sampling technique to select study subjects, we cannot say that the study population is representative of Malaysia. However, the study is representative of similar settings in Malaysia. We have added the above mentioned lines in the methods section. Please see the highlighted lines in the Methods section on page 8.

6. Page 15, line 3 from the bottom of paragraph: “….41.6% at the age group of 40-49 years…..” Instead of 41.6%, it was 41.8% written in Table 1.

Answer: This oversight has been corrected. Please see the highlighted percentage on page 15 of the Results section.

7. Page 24, line 4 from the top of paragraph: “Our results show the prevalence of MetS and its components to be comparatively lower among Chinese than in the Malays and the Indians, and this could be attributed to better lifestyle choices, including physical activity”. The sentence is overstated since there is not enough evidence to show Chinese have better lifestyle choices except the physical activity.

Answer: We agree with the reviewer’s comment and have revised the statement in this regard and have removed the words “lifestyle choices”. Please see the highlighted lines (lines 4-6) on page 24 of the revised manuscript.

8. The following references are similar studies and shall be mentioned in the paper:

- Kaur et al. (2018) Journal of Immigrant and Minority Health 20, 1380-1386.

- Narayanan et al. (2011) Metabolic Syndrome and Related Disorders 9, 389-395.

- Tan et al. (2011) Metabolic Syndrome and Related Disorders 9, 441-451.

Answer: These references have been incorporated in the manuscript. Please see reference number 29 (page 35), 40 and 43 (on page 37) in the reference section of the revised manuscript.

9. Some defining criteria of metabolic syndrome (page 12) are inaccurately written - the definitions of raised serum triglycerides and low HDL are missing the symbols of "equals to or more than"and "less than" respectively.

Answer: Thanks. These have been rectified. Please see the highlighted lines on page 12 of the revised manuscript.

10. Isolated grammatical errors are present including those listed below:

1)page 23; paragraph 2; line 1

2)page 24; paragraph 1; line 10

2) page 25; paragraph 2; line 6-7

Answer: Thanks.These have been rectified. Please see the highlighted lines on page 23-25 of the revised manuscript.

---

## [Editor Report · Decision Letter 2]

24 Feb 2020

Relationship of sociodemographic and lifestyle factors and diet habits with metabolic syndrome (MetS) among three ethnic groups of the Malaysian population

PONE-D-19-27618R2

Dear Dr. Iqbal,

We are pleased to inform you that your manuscript has been judged scientifically suitable for publication and will be formally accepted for publication once it complies with all outstanding technical requirements.

With kind regards,

Hoh Boon-Peng, PhD

Academic Editor

PLOS ONE
---

## [Editor Report · Acceptance letter]

2 Mar 2020

PONE-D-19-27618R2 

Relationship of sociodemographic and lifestyle factors and diet habits with metabolic syndrome (MetS) among three ethnic groups of the Malaysian population 

Dear Dr. Iqbal:

I am pleased to inform you that your manuscript has been deemed suitable for publication in PLOS ONE. Congratulations! Your manuscript is now with our production department. 

With kind regards,

on behalf of

Dr. Hoh Boon-Peng 

Academic Editor

PLOS ONE